# Accelerating Neural Differential Equations for Irregularly-Sampled Dynamical Systems using Variational Formulation

**Hongjue Zhao**
Department of Computer Science
University of Illinois at Urbana-Champaign
Champaign, IL 61820, USA
`hongjue2@illinois.edu`

**Yuchen Wang**
Department of Computer Science
William & Mary
Williamsburg, VA 23185, USA
`ywang142@wm.edu`

**Hairong Qi**
Department of Electrical Engineering and Computer Science
University of Tennessee, Knoxville
Knoxville, Tennessee 37996, USA
`hqi@utk.edu`

**Jiajia Li**
Department of Computer Science
North Carolina State University
Raleigh, NC 27695, USA
`jiajia.li@ncsu.edu`

**Lui Sha, Han Zhao**
Department of Computer Science
University of Illinois at Urbana-Champaign
Champaign, IL 61820, USA
`{lrs, hanzhao}@illinois.edu`

**Huajie Shao**
Department of Computer Science
William & Mary
Williamsburg, VA 23185, USA
`hshao@wm.edu`

## Abstract

Neural ODEs have exhibited remarkable capabilities in continuously modeling dynamical systems from observational data. However, existing training methods, often based on adaptive-step-size numerical ODE solvers, are time-consuming and may introduce additional errors. Despite recent attempts to address these issues, existing methods rely heavily on numerical ODE solvers and lack efficient solutions. In this work, we propose the Fast-VF Neural ODE, a novel approach based on variational formulation (VF) to accelerate the training of Neural ODEs for dynamical systems. To further mitigate the influence of oscillatory terms in the VF loss, we incorporate the Filon's method into our design. Extensive experimental results show that our method can accelerate the training of Neural ODEs by 10 $\times$ to 100 $\times$ compared to the baselines while achieving comparable accuracy in irregularly-sampled dynamical systems.

## 1 Introduction

Neural ordinary differential equations (Neural ODEs) (Chen et al., 2018) represent a family of continuous-depth machine learning models. Inspired by ResNets (He et al., 2016), Neural ODEs aim to parametrize vector fields of ODEs using neural networks,

$$\dot{x} = f_{\theta}(t, x), \tag{1}$$

where $f_{\theta} : [0, T] \times \mathbb{R}^d \to \mathbb{R}^d$ is a neural network and $\theta$ are model parameters. The continuous nature and special inductive bias of Neural ODEs make them pretty suitable to model dynamical systems from *irregularly-sampled time series* data. (Chen et al., 2018; Rubanova et al., 2019; Kidger et al., 2020). In existing training frameworks, numerical ODE solvers play a vital role. The forward pass results are directly calculated using numerical ODE solvers. In the backward pass, there are two major methods commonly employed to backpropagate through ODE solvers (Kidger, 2022; Onken & Ruthotto, 2020): (1) *discretize-then-optimize*, which involves directly backpropagating through operations of ODE solvers, and (2) *optimize-then-discretize*, also known as the *adjoint sensitivity*

*method*, as utilized in (Chen et al., 2018), which introduces additional adjoint ODEs. Details of these training methods are presented in Appendix A.

However, ODE solver-based training methods for Neural ODEs face two main limitations. First, they are inherently time-consuming. The internal mechanisms within numerical ODEs can incur significant computational costs in solving Neural ODEs. (Lehtimäki et al., 2022). This is attributed to the numerous evaluations of neural networks beyond desired time points. For the optimize-then-discretize approach, the case may worsen due to the additionally introduced adjoint ODEs. Secondly, existing approaches may suffer from low accuracy for two primary reasons. The auto-regressive nature of most numerical ODE solvers can lead to error accumulation. On the other hand, the optimize-then-discretize approach incurs additional numerical discretization error, resulting in inaccurate gradients, and potentially causing the training process to fail entirely (Gholami et al., 2019; Kidger, 2022).

Thus far, various approaches have been proposed to address the aforementioned limitations of Neural ODEs. However, these methods have not fully resolved these problems. To alleviate the computational bottleneck of Neural ODEs, some works attempt to constrain the complexity of learned dynamics, simplifying tasks for ODE solvers (Finlay et al., 2020; Kelly et al., 2020; Pal et al., 2021; Lehtimäki et al., 2022). Nevertheless, these approaches are unsuitable for time series tasks, where capturing the underlying dynamics is crucial. Moreover, they only accelerate the inference process of trained Neural ODEs, but the training process still remains time-consuming. Some studies also try to modify existing training methods directly (Daulbaev et al., 2020; Kidger et al., 2021; Djeumou et al., 2022; Norcliffe & Deisenroth, 2023). However, these approaches are still heavily dependent on the ODE solvers and the optimize-then-discretize technique, thus cannot effectively address the computational bottleneck. In certain cases, they may be slower than the discretize-then-optimize approach. Additionally, simulation-free training methods have emerged for continuous normalizing flows (Lipman et al., 2022; Ben-Hamu et al., 2022; Rozen et al., 2021). Despite their promise, these methods are not applicable to dynamical systems. In the domain of irregularly-sampled time series tasks, several models built upon neural differential equations have been proposed (Chen et al., 2018; Rubanova et al., 2019; Kidger et al., 2020). However, in essence, numerical ODE solvers are still the key components of these models, which potentially result in inaccuracies for long-term time series prediction.

To address these challenges, we strive to mitigate the usage of numerical ODE solvers during the training of Neural ODEs. To this end, we propose the Fast-VF Neural ODE, a novel approach for speeding up the training of Neural ODEs using the variational formulation (VF) (Hackbusch, 2017). Specifically, a loss function based on the VF (Qian et al., 2022) is employed in our proposed method. This VF loss only requires numerical integrations, thus neural networks are only needed to be evaluated on desired time points. Furthermore, we incorporate Filon's method (Deaño et al., 2017) into the loss function to address potential oscillatory integral issues effectively. We conduct extensive experiments to evaluate our method on various irregularly-sampled dynamical systems. Evaluation results show that our approach significantly outperforms existing baselines, achieving a $10\times$ to $100\times$ speed increase. In addition, it maintains higher or comparable accuracy levels.

In summary, our contributions include: (1) we developed Fast-VF Neural ODE, a novel training method using VF for accelerating Neural ODEs for dynamical systems; (2) we incorporate Filon's method to alleviate the influence of oscillatory terms in the VF loss for further enhancing model performance, (3) our approach achieves a significant acceleration, outperforming other baselines by 10 to 100 times in irregularly-sampled dynamical systems while achieving competitive accuracy.

## 2 METHOD

We first reviewed the basic concepts of VF and the Filon's method. Then, we elaborate on our proposed training framework designed to expedite Neural ODEs using these two techniques.

### 2.1 PRELIMINARIES

First of all, we review the VF of ODEs and Filon's method that will be used in our work.

**Variational Formulation (VF)** (Qian et al., 2022; Hackbusch, 2017). The variational formulation presented in Theorem 1 establishes a direct connection between the trajectory $\boldsymbol{x}$ and the vector field $\boldsymbol{f}$. This method enables Neural ODEs to learn parameters without relying on ODE solvers.

**Theorem 1.** *Consider $d \in \mathbb{N}^+$, $T \in \mathbb{R}^+$, continuous functions $\boldsymbol{x} : [0, T] \to \mathbb{R}^d$, $\boldsymbol{f} : [0, T] \times \mathbb{R}^d \to \mathbb{R}^d$, and $g \in \mathcal{C}^1[0, T]$, where $\mathcal{C}^1$ is the set of continuously differentiable functions. Here we define the functionals*

$$C_j(\boldsymbol{x}, \boldsymbol{f}, g) := \int_0^T [f_j(t, \boldsymbol{x}(t))g(t) + x_j(t)\dot{g}(t)]\mathrm{d}t, \quad j = 1, \dots, d. \tag{2}$$

*Then $\boldsymbol{x}$ is the solution to the system ODEs $\dot{\boldsymbol{x}} = \boldsymbol{f}(t, \boldsymbol{x})$ if and only if*

$$C_j(\boldsymbol{x}, \boldsymbol{f}, g) = 0, \quad \forall j \in \{1, \dots, d\}, \ \forall g \in \mathcal{C}^1[0, T] \ s.t. \ g(0) = g(T) = 0. \tag{3}$$

The proof of Theorem 1 can be found in (Qian et al., 2022).

**Filon's method for oscillatory integration.** Consider the integral $\int_a^b h(t) \sin(\omega t)\mathrm{d}t$ and suppose that only tabular data of $h(t)$ is available. As $\omega$ increases, the integrand will become oscillatory. In such case, it fails to perform Lagrange polynomial interpolation (Press, 2007) on the whole integrand using the general numerical integration techniques. To address this issue, Filon's method only performs Lagrange interpolation on $h(t)$, and then we can approximate $h(t)$ using a polynomial $p(t)$ with closed-form. Subsequently, the integral $\int_a^b p(t) \sin(\omega t)\mathrm{d}t$ can be directly computed to approximate the original integral. In this context, the polynomial is not required to approximate the oscillatory part in the integrand, thus effectively addressing the main problem that general numerical integration schemes encounter in oscillatory integration tasks. In real-world applications, Filon's method can be enhanced by dividing the integration interval into several segments, commonly known as the composite version of Filon's method.

## 2.2 PROPOSED METHOD

We detail our proposed Fast-VF Neural ODE. Given trajectories $\{\boldsymbol{x}^{[i]}(t)\}_{i=1}^N$ of the given dynamical system $\dot{\boldsymbol{x}} = \boldsymbol{f}(t, \boldsymbol{x})$. For the Neural ODE in Eq. (1), we aim to solve the following optimization problem to identify the optimal model parameters. According to (Qian et al., 2022), we have:

$$\boldsymbol{\theta}^* = \arg\min_{\boldsymbol{\theta}} \sum_{i=1}^N \sum_{j=1}^d \sum_{s=1}^S C_j^2(\boldsymbol{x}^{[i]}, \boldsymbol{f_\theta}, g_s), \tag{4}$$

where $C_j$ is defined in Equation (2). Since Theorem 1 specifies an infinite number of constraints, rendering it impractical for implementation, we employ $\{g_s\}_{s=1}^S$ to compute the loss function following (Qian et al., 2022). This set represents a collection of Hilbert orthonormal basis for $L^2[0, T]$, satisfying $\forall s \in \{1, \dots, S\}$, $g_s(0) = g_s(T) = 0$ and $g_s \in \mathcal{C}^1[0, T]$. In this work, we set $g_s(t) = \sqrt{2/T}\sin(s\pi t/T)$, which can be viewed as the Fourier basis. However, the Fourier basis introduces oscillatory terms into the VF loss. As $s$ increases, general numerical integration techniques will not work well, but this issue has not been addressed in (Qian et al., 2022). In our approach, we calculate $C_j$ numerically using the composite Filon's method, as described in Section 2.1, in which a second-degree polynomial is interpolated on each segment.

Next, we try to solve the optimization problem in Eq. (4). But the issue is that the functional $C_j$ in the VF loss is an integral from $0$ to $T$. However, in irregularly-sampled dynamical systems tasks, sampling time points may not begin at 0, for example, $[t_0, t_N]$. In this scenario, for autonomous systems, we can map the time interval $[t_0, t_N]$ to $[0, T]$ by defining $T = t_N - t_0$. Regarding non-autonomous systems, we can always transform them into autonomous systems by including $t$ as a new variable (Brunton et al., 2016). These adaptations allow our method to be applied effectively in irregularly-sampled dynamical systems.

Now we try to demonstrate how we achieve speed enhancement in our method, which comes from two aspects: (1) a reduction in the number of function evaluations (NFEs), and (2) improved parallelizability. Consider a trajectory sampled $K$ points. In our approach, the VF loss is calculated by evaluating the vector field exactly $K$ times in parallel to compute numerical integrals. In contrast, traditional ODE solver-based training methods often require evaluating the vector field more than

$K$ times step by step, as they typically need to assess the vector field at additional time points due to step size settings in numerical ODE solvers to maintain accuracy. Moreover, to perform one-step forward predictions in ODE solvers, the vector field must be evaluated multiple times in high-order ODE solvers such as `Dopri5`, a common choice in practice.

In summary, our method enables the learning of parameters in Neural ODEs without relying on numerical ODE solvers. Regarding computational efficiency, our approach requires only numerical integration for each trajectory. In these numerical integrals, vector fields of Neural ODEs are only evaluated at given time points in the dataset in parallel, significantly reducing the number of function evaluations and achieving better parallelizability compared with existing training methods. Additionally, our loss function effectively mitigates error accumulation from auto-regression.

## 3 EXPERIMENTS

We conduct extensive experiments to evaluate the proposed Fast-VF Neural ODE using irregularly-sampled data collected from various dynamical systems.

**Datasets.** We choose three dynamical systems pertinent to various fields such as biology, biochemistry, and genetics. These include the Gompertz model (Gompertz, 1825); the glycolytic oscillator (Sel'Kov, 1968); and the genetic toggle switch (Gardner et al., 2000). For each system, we first generate 125 trajectories with randomly sampled initial points. Among these trajectories, 100 trajectories are allocated for training, while the remaining 25 trajectories are used for validation. For these trajectories, we first sample them regularly at $0, \Delta t, 2\Delta t, \dots, T$. Subsequently, we randomly select points from the sampled points based on a specific ratio $r$. Additionally, 25 trajectories sampled at $0, \dots, 2T$ are generated for testing. Notably, the points at $0, \Delta t, \dots, T$ are utilized for interpolation tasks, while the points at $T, T + \Delta t, \dots, 2T$ are reserved for extrapolation tasks. In our experiments, $T$ is set to 10, and $\Delta t$ is set to 0.1. In addition, we choose $r = 0.8$ for our experiments. Details for datasets can be found in Appendix B.2.

**Baselines.** To evaluate the acceleration performance, we compare our method with the following training approaches: (1) the discretize-then-optimize approach (Dis-Opt), (2) the optimize-then-discretize approach (Opt-Dis) (Chen et al., 2018), and (3) the seminorm approach (Kidger et al., 2021). Additionally, we also compare our proposed method with following irregularly-sampled dynamical system tasks: (1) Vanilla Neural ODE (Chen et al., 2018), (2) Latent ODE with RNN encoder (Chen et al., 2018), (3) ODE-RNN (Rubanova et al., 2019), (4) Latent ODE with ODE-RNN encoder (Rubanova et al., 2019), (5) Neural CDEs (Kidger et al., 2020). Detailed training settings are presented in Appendix B.

**Evaluation Results.** We evaluate the performance of our method on interpolation and extrapolation tasks respectively. We evaluate the performance of different methods using the Mean Absolute Percentage Error MAPE $= \frac{1}{N} \sum_{i=1}^{N} \left| \frac{x_i - \hat{x}_i}{x_i} \right|$. To evaluate the acceleration performance of our method, we compute the training time according to prior works (Djeumou et al., 2022; Norcliffe & Deisenroth, 2023).

First, we compare the acceleration performance of our method against the baselines by considering an irregularly-sampling ratio of $r = 0.8$. As shown in Fig. 1, we can observe that our approach can speed up the training of Neural ODE by $10 \times$ to $100 \times$ compared to the baselines. In addition, we assess the mean absolute percent error across three dynamical systems for different methods, as illustrated in Table 1. The results indicate that the proposed approach can achieve competitive performance yet faster training speed than the best baseline ODE-RNN. Note that the good performance of ODE-RNN is attributed to its intricate network structure. In conclusion, our method can markedly accelerate model training while simultaneously maintaining high accuracy.

## 4 CONCLUSION

This work proposed a novel variational formulation-based training method to accelerate Neural ODEs for irregularly-sampled dynamical systems. Our method only required one numerical integration in the loss without the need of numerical ODE solvers. To address the potential oscillatory

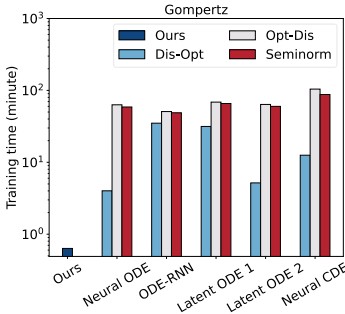 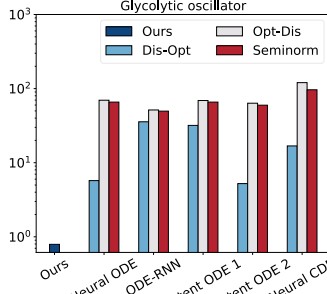 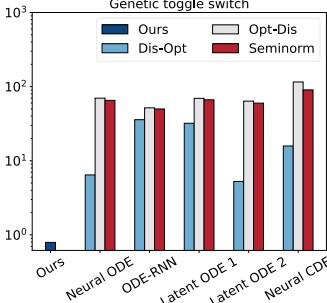

Figure 1: Training time (minute) for each method. We can see that our method can achieve $10\times$ to $100\times$ faster than the baselines. Note that Neural ODE denotes the Vanilla Neural ODE, Latent ODE 1 denotes the Latent ODE with RNN ecnoder, and Latent ODE 2 denotes the Latent ODE with ODE-RNN encoder.

Table 1: Testing Mean Absolute Percentage Error (MAPE) ($\times 10^{-2}$) on three dynamical systems with $r = 0.8$. We can see that our method can achieve comparable performance to other baselines.

| | Gompertz | | Toggle | | Glycolytic | |
|---|---|---|---|---|---|---|
| | Interp | Extrap | Interp | Extrap | Interp | Extrap |
| Vanilla Neural ODE | 0.430 | 0.1039 | NaN | NaN | 11.53 | 7.593 |
| ODE-RNN | **0.0829** | **0.0181** | 1.476 | 1.424 | **0.4593** | **0.1615** |
| Neural CDE | 0.513 | 4.60 | 6.397 | 109.3 | 1.969 | 33.39 |
| Latent ODE ( RNN Enc.) | 7.191 | 3.938 | 59.10 | 166.0 | 48.70 | 90.40 |
| Latent ODE ( ODE-RNN Enc.) | 5.856 | 8.289 | 37.85 | 218.9 | 16.46 | 78.94 |
| Ours | 0.089 | 0.053 | **0.398** | **0.332** | 0.556 | 0.843 |

integral challenge in the VF loss, we incorporated the Filon's method to enhance model performance. Evaluation results on three irregularly-sampled dynamical systems demonstrated that our method can significantly speed up the training of Neural ODEs while remaining high accuracy.

**Limitations of our method.** As our model relies on Filon's method, which uses polynomial interpolation for sampled trajectories, there are instances where polynomials may not effectively approximate complex trajectories. In such cases, the method's ability to capture intricate system dynamics might be compromised. In addition, extreme measurement settings for dynamical systems still remain as open challenges, e.g., extremely sparse noisy observations.

ACKNOWLEDGEMENTS

Research reported in this paper was sponsored in part by NSF under award CPS NSF-2311086 and Faculty Research Grant at William & Mary 141446.

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

## A    GENERAL TRAINING FRAMEWORKS OF NEURAL ODES

In this subsection, we conclude two primary training frameworks for Neural ODEs. Consider the Neural ODE expressed as Equation (1). In both training frameworks, the results of the forward pass are directly calculated numerically by ODE solvers, e.g., Runge-Kutta methods (Butcher, 2016):

$$\mathcal{L}(\boldsymbol{x}(t_1)) = \mathcal{L}\left(\boldsymbol{x}(t_0) + \int_{t_0}^{t_1} \boldsymbol{f_\theta}(t, \boldsymbol{x})\mathrm{d}t\right) = \mathcal{L}\left(\mathrm{ODESolve}(\boldsymbol{f_\theta}, \boldsymbol{x}(t_0), t_0, t_1)\right), \tag{5}$$

where $\mathcal{L}(\cdot)$ is the loss function. For the backward pass, there are two major approaches to calculate gradients: *discretize-then-optimize* and *optimize-then-discretize* (Kidger, 2022; Onken & Ruthotto, 2020). For the discretize-then-optimize approach, backpropagation is directly performed through the operations of ODE solvers. On the other hand, the *optimize-then-discretize* approach, also known as the *adjoint sensitivity method*, involves introducing backward-in-time adjoint ODEs,

$$\frac{\mathrm{d}\boldsymbol{a}}{\mathrm{d}t} = -\boldsymbol{a}^\top \frac{\partial \boldsymbol{f_\theta}(t, \boldsymbol{x})}{\partial \boldsymbol{x}}, \tag{6}$$

where $\boldsymbol{a} = \frac{\partial \mathcal{L}}{\partial \boldsymbol{z}}$ is the *adjoint state*. Subsequently, the gradients of the loss function $\mathcal{L}(\cdot)$ with respect to parameters $\boldsymbol{\theta}$ can be obtained by calculating a integral:

$$\frac{\partial \mathcal{L}}{\partial \boldsymbol{\theta}} = \int_{t_1}^{t_0} \boldsymbol{a}^\top \frac{\partial \boldsymbol{f_\theta}(t, \boldsymbol{x})}{\partial \boldsymbol{\theta}}\mathrm{d}t. \tag{7}$$

Above all, numerical ODE solvers play a pivotal role in training models based on neural differential equations.

## B    EXPERIMENTAL SETTINGS

### B.1    EXPERIMENTAL ENVIRONMENTS

All experiments in this work are implemented using JAX (Bradbury et al., 2018). Specifically, the implementation of neural differential equation models is based on Equinox (Kidger & Garcia, 2021) and Diffrax (Kidger, 2022). To optimize models, we use the Optax (DeepMind et al., 2020). All the experiments are implemented on the same server, equipped with a 40-Core 2.1 GHz Intel Xeon Gold 5218R CPU, 125GB of RAM, and two NVIDIA GeForce RTX 3090 GPUs, each with 24 GB of memory.

### B.2    SETTINGS OF DYNAMICAL SYSTEMS

In this subsection, we present the generation of datasets. In this work, we choose three dynamical systems, including the Gompertz model, glycolytic oscillator, and genetic toggle switch. The specific parameters for each dynamical system are provided as follows.

**Gompertz model.** The Gompertz model is widely applied in medical research and tumor growth analysis as a kind of growth model. It can be expressed as

$$\dot{x} = -\theta_1 x \cdot \log(\theta_2 x), \tag{8}$$

where $\theta_1 = \theta_2 = 1.5$, and $x(0) \in [0.1, 1.1]$.

**Glycolytic oscillator.** The glycolytic oscillator is a fundamental system in biochemistry that models the glycolysis process. It can be expressed as

$$\begin{aligned}
\dot{x}_1 &= \theta_1 - \theta_2 x_1 - x_1 x_2^2, \\
\dot{x}_2 &= -x_2 + \theta_3 x_1 + x_1 x_2^2,
\end{aligned} \tag{9}$$

where $\theta_1 = 0.75$, $\theta_2 = \theta_3 = 0.1$, and $x_1(0), x_2(0) \in [0.1, 1.1]$.

**Genetic toggle switch.** The genetic toggle switch is a key mechanism in genetic engineering and synthetic biology for controlling genes. It can be expressed as

$$\begin{aligned}
\dot{x}_1 &= \frac{a_1}{1 + x_2^{n_1}} - x_1, \\
\dot{x}_2 &= \frac{a_2}{1 + x_1^{n_2}} - x_2,
\end{aligned} \tag{10}$$

where $a_1 = a_2 = 4$, $n_1 = n_2 = 3$, and $x_1(0), x_2(0) \in [0.1, 4.0]$.

### B.3 TRAINING SETTINGS

In our experiments, each neural differential equation model is trained for 5000 epochs using the `Dopri5` solver. We employ the Adam optimizer (Kingma & Ba, 2014) with an initial learning rate of 0.001. The learning rate is scheduled using `cosine_onecycle_schedule`. All training data is loaded in one epoch. Data generation involves using the `Dopri5` solver in Diffrax for numerical ODE solutions. In the VF loss, we set the $S$ in Equation (4) to 100. For all RNN models throughout experiments, we use the Gated Recurrent Unit (GRU) (Cho et al., 2014). To ensure a fair comparison, the number of parameters across all models is set to be approximately equal. Results in Table 1 are obtained using the optimize-then-discretize approach, following (Kidger, 2022). With the same training settings, Vanilla Neural ODEs failed to learn the dynamics of the toggle switch based on `diffrax` proposed by (Kidger, 2022).

