# OpenReview forum: "Accelerating Neural Differential Equations for Irregularly-Sampled Dynamical Systems Using Variational Formulation"
_ICLR.cc/2024/Workshop/AI4DiffEqtnsInSci — AI4DiffEqtnsInSci @ ICLR 2024 Poster_

### Official Review · Reviewer_Gp8u · 2024-02-21
**Review of "Accelerating Neural Differential Equations for Irregularly-Sampled Dynamical Systems Using Variational Formulation"**

**Rating:** 4
**Confidence:** 4

**Review:**

The paper proposes a new training algorithm for neural ODEs based on a variational formulation. This effectively replaces the numerical ODE solver with numerical quadrature, with the goal of thereby accelerating the training procedure.

Overall, the proposed method seems interesting and promising, and I think it is a valuable addition to the neural ODE literature. Unfortunately, I have concerns regarding the presentation and in particular regarding some claims made by the authors, that seem overly strong and insufficiently supported. I will elaborate on individual points in the following.
- *Numerical ODE solvers as the bottleneck for neural ODEs:* This claim is made throughout the paper, in particular in the introduction and in the abstract, but I find this statement quite unclear.
  - "existing training method, often based on adaptive-step-size numerical ODE solvers, are time-consuming and may introduce additional errors": If anything, does step size adaptation not help with the efficiency by only evaluating the model when required? Also, in which sense does this introduce additional errors? Paragraph two states again that "ODE solver-based training methods [...] are inherently time-consuming", which again does not seem like a true statement.
  - "the optimize-then-discretize approach incurs additional numerical discretization error, resulting in inaccurate gradients, and potentially causng the training process to fail entirely": To the best of my knowledge, this is due to stability issues of the adjoint ODE, and can be mitigated e.g. with checkpointing or computing and storing an interpolant; see e.g. [1].
  - Paragraph 3 again states multiple times that ODE solvers are the computational bottleneck and leads to inaccuracies, without substantial support.
In summary, from the perspective of the initial Neural ODE paper by Chen et al, are ODE solvers not exactly the object that were added to ResNets that brought all of their nice benefits? An are adaptive steps not exactly the mechanism that saves computation? Of course, if there is a way to speed up the process that would be great! I just struggle to see the evidence for the claim that the paper tries to repeatedly make.
- *The computational cost of the proposed method:* The introduction mentions that "this VF loss requires one numerical integration" (similar statement at the end of section 2.2). But does equation 4 not show that $C_j$ needs to be computed for $S$ different $g_s$ functions, implying that we need to solve $S$ integrals? In any case, with the strong claims made in the paper this statement could and should be much more rigorous: How many evaluations of the ODE vector field, i.e. the neural network, do existing methods need, and how many are needed by the proposed method? These numbers are directly comparable, as the main assumption here seems to be that exactly thos evaluations are the expensive part I believe? Or if the speed-up comes from a different aspect, e.g. the better parallelizability of numerical quadrature vs numerical ODE solvers, then this should be elaborated on.
- *Insufficiently clear numerical evaluation:* The experiments compare many different methods, some of which are much more complicated than the simple neural ODEs that were required in the paper. I believe the evaluation would be strengthened by being much simpler. From my understanding, the main contribution of the paper is a novel loss function for training neural ODEs, as well as thereby also a novel way to compute parameter gradients. Just comparing this to existing methods would already provide helpful evidence. For example, benchmarking the cost of evaluating the loss function vs a standard loss function, and benchmarking the cost of evaluating gradients vs Opt-Dis and Dis-Opt and Seminorm; as well as also comparing the quality of both loss and gradients to some reference, e.g. computed by using an existing established ODE-solver based method with very low accuracy settings. A possible result of this would then hopefully be that all methods achieve similar quality of loss and gradients, but the proposed method presents a significant speed-up over the others. If then it was previously also discussed where exactly this comes from (number of NN evals or parallelisation?) then this would already be all the evidence that this paper needs. Of course, looking at multiple ODEs and even at ODE-RNNs etc is good additional evidence.

So overall, I do believe that the proposed method is valuable and should be presented to the neural ODE community, but in the current form the paper makes too many claims that are insufficiently supported. I would strongly recommend to the authors to revise the manuscript, soften many claims, and provide a simpler and clearer evaluation of the method, to better present this interesting method.


[1] Ma et al, "A Comparison of Automatic Differentiation and Continuous
Sensitivity Analysis for Derivatives of Differential Equation Solutions"

---

### Official Review · Reviewer_3nNo · 2024-02-27
**Review of "Accelerating Neural Differential Equations for Irregularly-Sampled Dynamical Systems Using Variational Formulation"**

**Rating:** 6
**Confidence:** 2

**Review:**

# Summary
The paper proposes a novel training method for neural differential equations based on the variational formulation (VF) of ODEs.
The benefit of the proposed method is that it bypasses the need for numerical ODE solvers during training, with the result that training may be accelerated by multiple orders of magnitude.
Additionally, the paper addresses the challenges of oscillatory integral terms in the proposed VF loss using Filon's method.
In a number of experiments, the proposed method is 1-2 orders of magnitude faster than the baseline architectures.

# Strengths
1. (**significance**)
The proposed method is well motivated and has the potential to solve a real problem with high impact across multiple scientific disciplines.
2. (**quality**)
The use of Filon's method to address the oscillatory integral problem is a key component of the paper's methodological contribution.
3. (**quality**, **clarity**)
The paper is well written and enjoyable to read.

# Weaknesses
1. (**clarity**)
The paper builds upon the work of Qian et al. [1], but I think a few more sentences would help to rigorously connect the two settings, which are slightly different (neural differential equations versus closed-form ODE discovery).
In particular, Qian et al. [1] go to great lengths to justify their training objective, an optimization problem over a function space.
The training objective of this paper (Eq. 4) is presented without comment as equivalent to Qian et al.'s.
After some consideration, I think the equivalence is probably valid.
However, I think the clarity of this paper would be improved if this step were justified more explicitly.
1. (**quality**, **completeness**)
The main issue I have with the paper in its current form is the choice of experiments.
To the best of my understanding, the "Fast-VF Neural ODE" is a standard, vanilla neural ODE architecture trained using the proposed VF method.
This is then compared to a number of more exotic neural ODE architectures, each of which is trained using three standard solver-based approaches (discretize-then-optimize, optimize-then-discretize, seminorm).
The problem is that each of these baselines is effectively "two steps removed" from Fast-VF Neural ODE, that is to say, both the architecture *and* the training method are different, meaning I cannot properly evaluate the effect of the proposed training method in isolation.
What I would like to see, at a minimum, is a comparison of Fast-VF Neural ODE with another vanilla neural ODE that was trained using each of the three solver-based approaches.
Conversely, if possible, it would also be interesting to see the more exotic architectures trained using the proposed VF method, which I think is possible in principle.

# Other Comments
1. A number of methods already exist for simulation-free training of neural ODEs in the context of continuous normalizing flows, where they parametrize probability paths; see, for example, [2-4].
I'm not sure whether these methods could be adapted to the present setting, but I think this paper would benefit from a discussion of them.
1. It seems to me that applying Filon's method is essential to making this method work.
With that in mind, I think future versions of this paper would benefit from a discussion of its limitations.
For example, are there situations where Filon's method might fail?

# Conclusion
Overall, I like this paper a lot.
Unfortunately, the lack of more meaningful and direct comparisons, as discussed in the second weakness above, prevents me from giving the paper a higher score at the moment.

# Citations
[1] Zhaozhi Qian, Krzysztof Kacprzyk, & Mihaela van der Schaar. D-CODE: Discovering Closed-form ODEs from Observed Trajectories. In *The Tenth International Conference on Learning Representations*, 2022.

[2] Yaron Lipman, Ricky T. Q. Chen, Heli Ben-Hamu, Maximilian Nickel, and Matthew Le. Flow Matching for Generative Modeling. In *The Eleventh International Conference on Learning Representations*, 2023.

[3] Heli Ben-Hamu, Samuel Cohen, Joey Bose, Brandon Amos, Maximillian Nickel, Aditya Grover, Ricky T. Q. Chen, Yaron Lipman. Matching Normalizing Flows and Probability Paths on Manifolds. In *Proceedings of the 39th International Conference on Machine Learning*, PMLR 162:1749-1763, 2022.

[4] Noam Rozen, Aditya Grover, Maximilian Nickel, Yaron Lipman. Moser Flow: Divergence-based Generative Modeling on Manifolds. In *Advances in Neural Information Processing Systems*, 2021.

---

### Meta-Review · Area_Chair_ReXw · 2024-02-28

**Recommendation:** Accept (Poster)

**Metareview:**

Dear Authors,

Thank you for submitting the draft.

Both reviewers agree that the presented work presents some interesting strengths. However, both reviewers do also raise some major points of concern, especially regarding the clarity of the presentation and some of the claims made. It is expected that authors will be addressing comments by the reviewers in the final draft.

regards

AC

---

### Decision · Program_Chairs · 2024-02-29

Accept (Poster)